# Parents’ Experiences of Direct and Indirect Implications of Sleep Quality on the Health of Children with ADHD: A Qualitative Study

**DOI:** 10.3390/ijerph192215099

**Published:** 2022-11-16

**Authors:** Ulrika Harris, Petra Svedberg, Katarina Aili, Jens M. Nygren, Ingrid Larsson

**Affiliations:** 1Blekinge Centre of Competence, SE-371 81 Karlskrona, Sweden; 2Department of Health and Care, School of Health and Welfare, Halmstad University, SE-301 18 Halmstad, Sweden; 3Department of Health and Sport, School of Health and Welfare, Halmstad University, SE-301 18 Halmstad, Sweden

**Keywords:** ADHD, children, parents, sleep, ability, well-being, qualitative content analysis, abductive approach

## Abstract

Sleep problems represent a significant challenge for children with ADHD. However, lack of knowledge about how sleep affects children with ADHD in terms of their health and everyday life prevents the development and implementation of interventions to promote sleep. The aim of this study was to explore parents’ experiences of direct and indirect implications of sleep quality on the health of children with ADHD. The study used an abductive qualitative design, with Tengland’s two-dimensional theory of health as a deductive analysis framework. Semi-structured interviews were conducted with 21 parents of children aged 6–13 with ADHD and sleep problems. The parents experienced that sleep influenced their children’s abilities to control emotional behaviour related to ADHD and to manage everyday life. Sleep also had an impact on the children’s well-being, in relation to both vitality and self-esteem. In conclusion, the results show important direct and indirect implications of sleep quality on the health of children with ADHD. This implies a need for greater focus on sleep, to target both abilities and well-being in promoting health among children with ADHD.

## 1. Introduction

Sleep is fundamental to general health and well-being, vital for physical and mental recovery [1], and crucial in strengthening learning and memory [2]. Sleep problems can contribute to stress, weakening of immune responses, and disrupted functioning in daily life [1]. Good sleep quality is especially important for children and adolescents, and can have a positive impact on their sense of optimism, self-esteem, and general well-being [3]. Long-term sleep problems can have a negative impact on children’s development, daily functioning, social relations, academic performance, and quality of life [4,5,6]. Parents are also affected by their children’s sleep problems, with increased stress levels, disturbed sleep, and general mental health issues [7]. Compared to children in the general population, sleeping problems are particularly prominent among children with ADHD, and the consequences of the problems are more severe. Such negative consequences include decreased abilities to control both impulses and anger, decreased ability to concentrate, and, in the long-term, reduced academic performances, social exclusion, and health issues [8].

About 50–70% of children with ADHD suffer from some sort of sleep problems [9,10], compared to 40% of children without ADHD [11]. The type of sleep disorders occurring among children with ADHD can be divided into insomnias, sleep-related breathing problems (such as obstructive sleep apnoea), and parasomnias. Sleep onset insomnia and sleep maintenance insomnia, including problems with refusing to go to bed, difficulty going to sleep, waking up several times at night, and snoring, nightmares, and daytime sleepiness, are the most frequently reported problems among children with ADHD [12]. A study comparing various subtypes of ADHD shows that children with primarily hyperactive-impulsive ADHD have the most sleep problems, specifically insomnia and nightmares [4]. The cause of sleep problems in children with ADHD is complex and not easily defined. ADHD stimulant medication such as methylphenidate is one explanation for sleep problems [13]. Other explanations encompass family or environmental factors, or neurobiological origins [7]. Some studies emphasize lifestyle factors, such as a high-energy diet (fat, carbohydrates, and sugar), frequent snacking, a sedentary lifestyle, overuse of digital devices and gaming, and delayed bedtimes [14,15,16]. During the COVID-19 pandemic, other challenges have emerged for families in relation to maintaining good sleep routines. Lockdowns and remote learning have resulted in deprivation of leisure activities, disturbed daily routines, and, ultimately, in increased stress and anxiety for families already struggling to maintain good sleep routines [17]. The Swedish response to the COVID-19 pandemic was different from many other countries since it did not include lockdowns. Schools were open as usual, only upper secondary schools (age 13–18) used remote learning for a shorter period of time [18].

Sleep problems in children with ADHD are commonly treated with melatonin [19], a hormone that regulates the circadian rhythm and creates a feeling of sleepiness. It is considered to be a safe and reliable way to treat sleep problems in children [20]. Sleep problems can also be treated with non-pharmacological interventions, such as education on sleep hygiene, which is aimed at parents [21], cognitive behavioural therapy aimed at children [22], or weighted blankets [23]. Considering the complexity of sleep problems within this target group, it is important to acknowledge various perspectives, including those of children and their families, to understand the potential impact of pharmacological and non-pharmacological interventions that target children with ADHD. There is a lack of qualitative studies exploring parents’ experiences of how sleep affects children with ADHD in relation to their health and everyday life [8]. Given that parents live in close connection with their children, they have a profound knowledge of their daily life and how their sleep affects them. This highlights the need to understand parents’ perspectives in order to develop and implement interventions (which could be based on pharmacological and non-pharmacological treatment) that could positively impact their children’s health. Despite the importance of sleep, it tends not to be prioritised in health care treatment for children with ADHD [24]. From a clinical perspective, focusing more on sleep could benefit the well-being and overall functioning of the child and the whole family [5]. In order to gain more knowledge on the different perspectives related to the complexity of sleep and how it influences the abilities and well-being of children with ADHD, this qualitative study aimed to explore parents’ various experiences of direct and indirect implications of sleep quality on the health of children with ADHD.

## 2. Materials and Methods

### 2.1. Study Design

The study had an explorative design, using qualitative content analysis, a qualitative method with a focus on subjective experience, the context, and variations of experiences. Qualitative content analysis focuses on similarities and differences in the data, being both descriptive (positivistic paradigm) and interpretative (hermeneutic paradigm). The study used an abductive approach, meaning that the data analysis moved back and forth between inductive and deductive approaches [25]. The deductive approach was based on the two-dimensional theory of health by Tengland, which takes a holistic view of health and includes the “ability” and “well-being” dimensions [26]. To ensure trustworthiness, the study complied with the consolidated criteria for reporting qualitative research 32-item checklist [27].

### 2.2. Participants

The participants were recruited from a sleep intervention project that included children with newly diagnosed ADHD and sleep problems (for more information, see [28]). A purposeful sample of parents were asked to participate in this interview study. The intention was to reach a variety of parents, in terms of sex, age, civil status, education, employment, and place of residence. The inclusion criteria were parents of a child aged 6–13 with ADHD and sleep problems. Parents with insufficient Swedish to take part in an interview were excluded. Initially, 23 parents of children with ADHD were approached with a request to participate in the study; of these, 21 parents agreed to participate (Table 1).

### 2.3. Data Collection

The interviews were conducted in person at the university by the last author (IL), from March to September 2020, during the first wave of the COVID-19 pandemic. The parents were given the option of a digital interview, but everyone chose to be interviewed in person at the university. The interviews took place at a large table with a recommended distance between participants. Interviews were performed with 17 parents individually, according to their choice, while four parents chose to be interviewed in pairs. All interviews were held in Swedish. The participants were asked questions on how they perceived their children’s sleep, what influences the children’s sleep, how sleep influences the various parts of the day for the children, their well-being, ADHD symptoms, and how the family is affected by the children’s sleep problems. Follow-up and probing questions, such as “Please, tell me more about…” were used when required, to further encourage the parents in their responses. A pilot interview was conducted in order to refine the interview questions. Since none of the interview questions needed adjustment, the pilot interview was included in the study. The interviews lasted between 41–64 min (median 53 min). All interviews were digitally recorded and transcribed verbatim.

### 2.4. Data Analysis

The interviews were analysed by using qualitative content analysis with an abductive approach, in accordance with Graneheim et al. [25]. The analysis followed several steps. In the first step, an inductive approach was taken, by reading the entire material (unit of analysis) several times. In the second step, the focus was on searching for variations, similarities, and differences in the interviews, and 178 meaning units related to the aim of the study were identified. In the third step, the meaning units were then condensed, coded and inductively grouped into eight preliminary subcategories with a focus on the manifest content. The eight preliminary subcategories were then merged into four subcategories. For example, “functioning in school” and “functioning in everyday life” were merged into the subcategory *managing everyday life*, and “handle temper” and “handle ADHD symptoms” were merged into the subcategory *controlling emotional behaviour*. In the fourth step, a deductive approach was taken to further develop the four subcategories and compare them to Tengland’s [26] two-dimensional theory of health and the dimensions of “ability” and “well-being”. The “ability” dimension is defined as an observable form of action and consists of an individual’s abilities, dispositions, motivation, and mental states. It is described as involving all that a healthy person needs in his/her context to be able to act and perform according to society’s norms. The dimension of “well-being” is defined as a subjective experience of a feeling. It concerns one’s mood and sensations, and could include, for example, feeling vital, energetic, calm, strong, or concentrated [26]. In the fifth and final deductive step of the data analysis, the process was iterative—continually reviewing the four subcategories, by moving from the empirical material to Tengland’s theory and then back again. This process was done to ensure that the theory was true to the core content of the interviews, and that no important data were left out. This step resulted in the theoretical model’s two dimensions (ability and well-being) forming the two categories, and the inductively derived four subcategories were divided into the relevant category. Thus, *controlling emotional behaviour* (66 meaning units) and *managing everyday life* (53 meaning units) were in the “ability” category, and *feeling vital* (36 meaning units) and *experiencing self-esteem* (23 meaning units) were in the “well-being” category. The content of the subcategories contributed with confirmed and new aspects of the dimensions of the theory, thereby broadening the understanding of how sleep influences the health (ability and well-being) of children with ADHD. Three authors (UH, IL, and PS) were involved in the entire analysis process, thus enabling a variety of interpretations of the material before consensus was reached among all authors [25].

### 2.5. Ethical Considerations

Ethical approval was granted by the Swedish Ethical Review Authority (no. 2019-02158) and followed the principles of the Declaration of Helsinki [29]. According to the Swedish Research Council [30], all research must comply with guidelines to ensure safety and confidentiality for the participants, by following ethical principles of autonomy, beneficence, non-maleficence, and justice. All participants provided their informed consent after receiving the researcher’s written and oral information about the study. Participation was voluntary, with the option of withdrawing from the study at any time without giving a reason.

## 3. Results

The parents experienced that sleep quality had direct and indirect implications on their children’s abilities and well-being. Related to abilities, after having a good sleep during the night, the children had better control over their emotional behaviour and management of their everyday life. The parents also experienced that their children could more easily feel vital and experience self-esteem through sleeping well, contributing to improved well-being. It emerged that the ability and well-being dimensions were closely interrelated, and are equally important and necessary to a description of how sleep quality influences what constitutes health for children with ADHD (See Figure 1).

### 3.1. Ability

The parents described that their children’s abilities to control their emotional behaviour and to manage everyday life improved after a good night’s sleep. In Tengland’s [25] model, the ability dimension is an observable action and involves all that a healthy person needs in his/her context to be able to act and perform according to the norms in society. The parents observed that sleeping well increased their children’s motivation and ability to act in accordance with norms, especially at school. Further, the parents experienced that the children were able to learn new things, feel competent, and develop as a person, all essential factors in becoming a healthy, well-functioning person.

#### 3.1.1. Controlling Emotional Behaviour

Having a good night’s sleep meant that the children were able to control anger and impulses to a greater extent and handle stress and difficulties when they encountered them in their daily environment. This occurred both at home and school, leading to more harmony and balance in the family and fewer conflicts with peers. The parents experienced that their children coped with difficulties in a much calmer way and were more patient, thus avoiding an escalation of anger that could lead to outbursts. According to the parents, lack of sleep intensified ADHD symptoms, with increased hyperactivity, more angry outbursts, lack of concentration, and less impulse control.


*If she hasn’t slept well, you notice that she’s more tired and grumpy. Not as cooperative. She doesn’t want to do as we say. But then she’s very, very stubborn in any case, most of the time. It gets 10 times worse if she hasn’t slept properly.*
(Parent no. 7)

Parents described how sleeping well led to less angry outbursts and conflicts. The children were able to handle setbacks and disappointments to a greater degree. Small problems stayed small, instead of growing out of proportion and affecting the child’s whole day. When a child had decent sleep, family life no longer revolved around preparing for and managing conflicts, resulting in greater harmony at home. The whole family, both children and adults, experienced a greater “flow” in the days.


*The days go by. It’s not a constant battle all the time about what to do or any argument; rather, you complement each other more at home, including the children. NN can be involved in activities that one has planned for oneself, it’s not just focusing and planning around him, because that’s how it was for a while. You prepared everything so it wouldn’t turn into a conflict or so he wouldn’t get angry. Now that he can handle setbacks, it’s easier to slow down, oneself, and take it a bit more as it comes.*
(Parent no. 21)

By being able to control their temper and avoid negative situations, the children were able to feel good about themselves and to feel a sense of control over their emotions and actions. The parents expressed how loss of control made the children feel worse about themselves. When the children acted in a way that they did not want to act, it created more negative feelings.


*I think that it (sleep) means a lot, I think that with poorer sleep he would have been angrier, and when he’s angrier he gets more hyperactive, and then he gets unsettled and then he’ll do something stupid, and then it creates a bit of anxiety, so I think it has a great impact.*
(Parent no. 17)

The ability to consider other people’s needs and feel empathy was also affected by sleep quality. Some parents noticed that the child interacted with and responded to other people and their needs depending on their level of tiredness.


*She loses touch with reality completely. It’s all about her at that moment, everything is focused on her. She can’t consider anyone else in that situation, it’s just panic all over. It’s obvious when you’re looking at her.*
(Parent no. 10)

#### 3.1.2. Managing Everyday Life

Parents described that their children could manage everyday life when they were well rested. This meant that the children could get up in the morning, get ready, take care of basic hygiene, prepare and eat meals, do homework, and communicate. Sleeping well also influenced the children’s functioning at school; they could focus, interact with peers, and cope with the school environment to a greater degree. The general motivation of the children increased with a good night’s sleep and enhanced their ability to learn, perform, and gain competence. The parents experienced that, by managing everyday life, their children had a better chance of developing and doing the things they wanted to do. The children were able to receive new ideas, be spontaneous, help out at home, and be more independent.


*You can do more things together because he has more strength, more energy and a greater desire to do things. A lot of the time he wants to come along when you’re going somewhere. I can’t take more than a few steps outside the door before “Mum, I want to come along”. He wants to help out more. It used to be a burden having to do a lot of things by ourselves. Now we get a break when he wants to help out. Come along to the shop and carry groceries and things like that.*
(Parent no. 15)

After a bad night’s sleep, the children struggled to stay focused at school. Often, the school informed the parents when this happened, and sometimes the parents could predict in the morning whether the school day would turn out positive or negative, based on their children’s behaviour and attitude. The parents described how they tried to create the right conditions for their children by helping them get better sleep. Through this, the children were able to manage everyday life, primarily at school, but also at home.


*Even if he is unsettled the next day, he would probably have been even more worried if he hadn’t had any sleep. So, it still feels like you’re helping, helping him with mental recovery and balance, as much balance that you can have when you’re seven and have ADHD, but it becomes a tool that can help him to function better.*
(Parent no. 17)

Even if sleep was not the only factor that controlled their children’s ability to manage everyday life, the parents recognised sleep as a crucial factor. Some parents had tried various strategies to help their children to get good sleep and, with time, discovered what worked and what did not work. However, they also acknowledged that the complexity of sleep meant that it was hard to understand how all the various factors affected each other.


*I think that he would be able to manage the surrounding world. Have better margins, and better ability to concentrate with better sleep. I’m thinking that it’s connected to learning as well. I think sleep is really important. So, all the bits of the puzzle, so to speak. You must combine all the factors to make it work.*
(Parent no. 2)

### 3.2. Well-Being

The parents experienced that sleeping well led to increased well-being for the children. This involved feeling vital and having self-esteem. In relation to Tengland’s [25] definition of well-being, i.e., concerning the mood and sensations, the parents sensed that the mood, in the sense of both the ambience and the children’s state of mind, improved with a good night’s sleep. One aspect that emerges in the results, but which does not appear strongly in Tengland’s theory, is the increased zest for life in the children that came with good sleep. This was evident in different ways, both in general but also in specific behaviour observed by the parents.

#### 3.2.1. Feeling Vital

The parents described how a good night’s sleep contributed to their children having greater energy and vitality. This transpired in undertaking leisure activities, having a social life with friends, spending time with family, and feeling greater harmony in the family. Sleeping well aided the children to play, use their imagination, and be more active. They had the energy to act on spontaneous ideas and explore where the ideas led them. The children could enjoy life more, compared to when they were feeling tired, resulting in greater well-being.


*He falls asleep at a reasonable time and gets up by himself in the morning and sort of wakes me up and asks when he has to go to school. He thinks that school was fun today. Yesterday he said that school is fun. One can tell that something has changed.*
(Parent no. 12)

Not getting enough sleep meant that the children had a more difficult time. It became harder to do what was expected of them. Sleeping well led to a more stable energy level throughout the day, instead of crashing at home straight after school. Overall, the parents perceived that this led to greater well-being.


*I think that, at school, she holds it all together, as well as she can. But when she gets home, then she’s ready to drop. So it’s mainly at home that it will come out. But I definitely think it has an effect, everything is much harder in a way. With concentration and holding it together and interacting with friends, everything gets so much harder.*
(Parent no. 8)

For some parents, when the children slept well, there was a radical change. It was as if a new child emerged. The parents sensed that there was a different interaction with the children.


*You can sit and talk to him in different way, just like he has woken up, somehow. Matured, probably, it feels like that. But sensing that sleep affects so much as a whole, both brain and body. That he is present in a different way.*
(Parent no. 3)

To have energy to hang out with friends after school was something that the parents noticed improved their children’s lives. The increased social life meant that the children could feel more included, leading to greater well-being.


*[She] started to hang out so much more with friends. Before it was more like, after school, she just wanted to lie in bed and take it easy and play a little bit. But now she hardly has time to sit still, she wants to do things.*
(Parent no. 9)

The parents also observed that physical activity led to better sleep and talked about how the two influenced each other. Physical activity was a factor that the parents found important in relation to their children’s well-being. Feeling more vital and having more energy resulted in increased physical activity in the children.


*But if you’re not really tired, then it’s easy for your thoughts to wander, it spins and spins and you keep thinking, then you can’t really sleep. My experience is that if you exhaust yourself properly and let the body work regularly you will fall asleep easier, it will happen quicker, and you’ll sleep better. It’s a win-win.*
(Parent no. 19)

Sleeping well and having night-time routines resulted in decreased mental health issues, which some parents expressed that their children suffered from, such as worry, anxiety, and ruminating thoughts. When children slept well, it led to them being calmer, happier, and able to feel better about themselves. Parents experienced a paradox in sensing a greater calmness in their children, while at the same time sensing they had more energy and vitality.


*I can see above all that his worry is…that is what I’m basing this on. That it’s such a difference. […] It’s gone. It’s kind of, well, almost completely gone all day, I can say. Otherwise, he can be, not what I would call worried, but energy stressed. But this worrying over everything.*
(Parent no. 3)

For some parents, putting the child to bed had been a real struggle, comparing it to a nightly battle. This constant struggle, which was especially noticeable in relation to the younger children, resulted in a great amount of stress and lack of recovery for the parents. The stress and exhaustion in the parents in relation to this issue led to even more stress about how it was affecting the child’s overall well-being. For some parents, it felt like taking part in a lottery, trying to figure out how to win the prize of a good night’s sleep for their child.


*He used to move around a lot in the bed, and then he was completely knackered at night, and then we had a messy night. And then we had that cycle every day, every night.*
(Parent no. 17)

#### 3.2.2. Experiencing Self-Esteem

Sleep problems meant that the children couldn’t perform as well or set and reach their goals as much, and, therefore, did not gain self-confidence. This, in turn, led to decreased self-esteem and well-being. The parents experienced that a feeling of failure was common in their children, especially in the school context, where they were being constantly compared to their peers and where the children were aware that they could not concentrate and function in the same way as the others, in the expected way. With good sleep, the children could succeed more.


*But it’s not that many days now that you get the call from school anymore. […]. It used to be more often and then it was really, we have even sat down with the school, and what are we going to do so she can cope all day.*
(Parent no. 5)

When they slept well, the children felt self-confident in the home environment and during the leisure activities. The parents experienced that a good night’s sleep helped the children finish what they had started, focus properly, reflect on issues, and solve problems without being controlled by their impulses. This meant that the children could feel better about themselves for achieving goals and feel greater self-esteem, which contributed to increased well-being.


*Yes, he loves to play with Lego, for example. After getting a good night’s sleep he can sit by himself for ages, but if he’s tired or hungry, then he gets grumpy over the slightest thing. He has a harder time understanding the instructions and finds it harder to focus. You can see, when he’s alert, that he thinks things over before letting the feelings come up.*
(Parent no. 14)

The parents described the importance of the children feeling included and “normal” by being able to do what their peers were doing at school, at play and leisure time, and participate in everyday life. As an example, some parents mentioned that, because their children struggled to learn to fall asleep by themselves, or woke up several times at night, they therefore felt too embarrassed to have sleepovers. This aspect of their sleep problems also affected the children’s self-esteem.


*But one thing that has become problematic is that everyone is having sleepovers more often. He wanted to have a sleepover but didn’t manage to do it. It has become difficult for him. It has really been driving this, that we’re now trying to get him closer to being able to fall asleep by himself.*
(Parent no. 4)

## 4. Discussion

This study presents parents’ experiences of direct and indirect implications of sleep on the health of children with ADHD, using Tengland’s theoretical framework, in which “abilities” and “well-being” form the two dimensions of health. To the best of our knowledge, this is the first qualitative study investigating parents’ views of how sleep in children with ADHD influences their health. The principal findings showed that good sleep quality increased the children’s ability to control their emotional behaviour, manage everyday life, improve their focus, control impulses and do the things they wanted to do, thereby impacting their ability to learn and develop. Another principal finding was that the children’s well-being increased through sleeping well; they felt more vital and energetic and had greater self-esteem and satisfaction in life. The parents described that good sleep quality allowed their children to be active, form new ideas, be spontaneous, socialise with family and friends, and complete things they started. The parents also experienced that mental health issues such as worrying, anxiety, and ruminating thoughts decreased through sleeping well. Altogether, the results revealed that good sleep quality influences both the ability and well-being dimensions of health among children with ADHD [26]. Further, the results highlight the complexity of defining health, given that the two dimensions are closely connected and influence each other in an ongoing process.

### 4.1. Ability

The parents expressed that, after sleeping well, their children could handle disappointments, control their anger better, and manage ADHD symptoms, such as restlessness and lack of concentration. Research shows that there are bidirectional associations between sleep problems and emotional dysregulation [31]. For children with ADHD, it is important to manage both negative and positive emotions, such as anger and overexcitement, because the lack of control can negatively affect social and functional outcomes [32]. The parents in the present study experienced that their children had peer problems due to their difficulty in controlling their emotions. Positive peer functioning is important, because children learn cooperation and negotiation and develop together with their peers [33]. By being rejected by the preferred peers repeatedly in childhood, there is a risk in adolescence of accepting friendships from peers with negative associations. Peer problems can thus lead to negative outcomes later in life, such as dropping out of school and substance abuse [34,35]. The parents in the present study experienced that good sleep increased their children’s ability to control their temper and cope with setbacks, which reduced the risk of conflicts. The ability to control emotional behaviour can thus have, to some extent, an effect on peer functioning and on both short-term and long-term consequences in life [32].

The present findings show that good sleep quality led to a greater “flow” in the family’s life, since the focus was no longer on managing their children’s temper and conflicts, resulting in more harmony at home. Previous studies of 10-year-olds and adolescents with ADHD suggest that better sleep improves parent-reported functioning and behaviour at home [36,37]. The parents experienced that, after sleeping well, the children could communicate in an easier way and problems did not escalate as they would otherwise. The parents further noticed that their children’s empathy seemed to increase when they slept well and that they had improved communication. This is particularly relevant in families with children with ADHD, given that family relationships can be strained due to poorer social and communication skills [38], and due to empathy difficulties in the children [39].

The results show that children’s ability to manage everyday life was also affected by sleep. The parents described how a good night’s sleep increased their children’s ability to get up in the morning, get ready for the day and be active throughout the day. The children’s ability to function at school also improved when they slept well and stimulated learning, focusing, and managing the school environment. Overall, the children’s general motivation increased, which enhanced their ability to learn, perform, and gain competence at school and home. Research has shown that both ADHD symptoms [40] and poor sleep [2] negatively affect children’s performance in school. Further, sleep problems can cause poor academic results, linked to daytime sleepiness, decreased executive functioning, and declined general health and motivation [2]. It is also essential to improve sleep among children at an early stage, given that sleep problems that persist over time cause reduced academic performance [6]. For children with ADHD and sleep problems, there is a high risk of failing academically, with potentially lifelong consequences [40]. The results from the present study underscore the need for these children to obtain good sleep, to give them the prerequisites necessary to perform academically and gain confidence.

### 4.2. Well-Being

The results show that children had more energy and felt vital when they slept well, leading to increased well-being. The parents sensed that good sleep quality resulted in greater harmony in the child, themselves, and the whole family. The children’s new energy levels led to more enjoyment in life, through playing, trying new things, coming up with new ideas, and being spontaneous. The parents also perceived that their children had a more stable energy level throughout the day. The findings from our qualitative study add value to a previous quantitative study revealing similar results, in which sleep problems were found to severely decrease quality of life in children with ADHD, in relation to physical and emotional well-being, self-esteem, family, friends, and school [4].

By getting a good night’s sleep, the parents experienced that their children could focus more in the classroom, like their peers, and could participate in playing and leisure activities, leading to greater self-confidence and feeling “normal”. The parents perceived that a feeling of failure was common in the children, especially in relation to the school context. At school, the children were expected to behave in a certain way, and they struggled with this to a great extent, compared to their peers, and even more so when they had sleep problems. This could influence well-being later in life, because it is common for adults with ADHD to live in a disappointment loop, due to their struggles to cope with expectations and feeling a lack of control, thus leading to reduced self-esteem [41]. By not living up to expectations, the children in the present study had less chance of performing and learning, and less chance of setting and reaching their goals. It also resulted in feeling different to their peers at school, leading to a feeling of exclusion. Good sleep quality enabled the children to live up to expectations and feel more included, which are fundamental to children’s well-being [42].

Although the findings from the present study point to sleep as the essential element in the children’s daily life and well-being, the results are not consistent. For example, various things can cause positive and negative aspects in children’s life: ADHD symptoms, lack of sleep, school stress, or a particular developmental phase in the child were some things mentioned by the parents. Additionally, in the present study, some children were going through a phase with many thoughts, worries, conflicts with friends, and mood swings, which are associated with being or becoming a teenager. These factors are common in comorbidity of psychiatric conditions in children with ADHD, and there are consistent links between anxiety or depression and sleep problems, and a bidirectional relationship between emotional problems and sleep problems [8]. In the present study, some of the children had problems with anxiety and ruminating thoughts, and the parents experienced that a good night’s sleep decreased these symptoms. Given that anxiety is common in children with ADHD [43], this further stresses the importance of sleep interventions.

In the present study, the parents experienced that the children showed signs of having negative attitudes about themselves. Helping their children to sleep well became a way of enabling the children to succeed, leading to greater self-esteem and well-being. Self-esteem can be defined as having positive or negative attitudes about oneself, often deeply rooted. Low self-esteem comes from negative experiences, such as living in an unsupportive environment, judging yourself, or being afraid of failure [41]. The children in the present study showed difficulties related to social, emotional, and academic factors, and experienced and expected failure in some or all of them. Good sleep quality increased the children’s chances of success. Thus, it is essential to improve children’s sleep, given that good sleep has been found to increase optimism, self-esteem, and general well-being in eight-year-olds [3]. Higher self-esteem is further associated with healthy choices, and low self-esteem in adolescent girls linked to family and school has been found to increase health risk behaviour, such as drug abuse and risky sexual behaviour [44]. The results from the present study are important findings and show a great need to offer children with ADHD and sleep problems sleep interventions to avoid long-term negative outcomes related to self-esteem and well-being.

### 4.3. Methodological Considerations

Trustworthiness in qualitative research is usually defined with the four criteria of credibility, dependability, confirmability, and transferability [25,45]. Credibility refers to how credible the data is, meaning that the study measures what it was supposed to measure, with credible interpretation of the results and a focus on the phenomenon explored. A possible weakness of this study is that only a few fathers participated, though there was a variation among the participants in other aspects—related to age, education level and the age of the child—which strengthens the study’s credibility [45]. The study included children aged 6–13. The age-related differences in sleep behaviours could have an impact on parents’ perceptions. This is a limitation, but at the same time a strength, since a range of different experiences are captured, and can be compared to other studies featuring children with ADHD and sleep problems of an even broader age range [10].

Most of the interviews were conducted individually, and two were conducted in couples, according to the parents’ preferences. Even though this decreased the consistency in the method, the couple interviews gave the parents the chance to reflect together, which deepened their reasoning. Credibility was further ensured by carefully describing the data collection and analysis, and through the ongoing discussion between the researchers in the analysis process. The researcher performing the interviews (IL) has extensive experience in qualitative methodology.

Dependability relates to stability of the data over time. All the participants in the study were given the same questions, follow-up questions were offered to avoid misinterpretations, and the participants were encouraged to talk freely. Another strength is that the same researcher conducted all the interviews in the same location. A possible weakness is that the interviewed parents were not offered the chance to read through the transcript interviews in order to correct any potential misunderstandings [27]. Dependability can also be affected due to the fact that the interviews were conducted during the first wave of the COVID-19 pandemic. However, Sweden did not have lockdowns that many other countries had. In Sweden, lower secondary schools were open as usual. Upper secondary schools (age 13–18) did have remote learning for a limited time. School activities and leisure activities continued with some adjustments to prevent infection spreading [18]. Even though the situation in Sweden differed from other countries with more restrictions, it is important to consider how the pandemic could have affected the experiences of the children and parents in this study, and thus the dependability.

Confirmability refers to objectivity in relation to how much the researchers influenced the results. Systematic data analysis was ensured by using the recommended steps in qualitative content analysis, as reported earlier in this article. Three researchers with different backgrounds and experiences (UH, IL, and PS) were involved in the whole analysis process, enabling diverse interpretations of the material before reaching a consensus [25]. Transferability refers to the results being transferable to similar contexts. Considering that the participants were selected in a clinical context with careful diagnosis and selection, the results can most likely be transferred to other children with ADHD and their families. A potential limitation is that all the participants were recruited from the same clinic. Transferability can be reached through a “thick description” [45] of the participants and the results. The results, therefore, contain numerous quotations, to further illuminate and let the reader follow the data analysis process.

## 5. Conclusions

The results show important direct and indirect implications of sleep quality on the health of children with ADHD. According to the parents’ experiences, sleep influences the health of children with ADHD to a great extent. The children’s abilities to control their emotional behaviour and manage everyday life were improved with a good night’s sleep. The children felt more vital and had better self-esteem due to sleeping well, resulting in increased well-being. This result could imply that a greater focus on sleep is needed in a clinical context, involving children and their families, to ensure that individual needs and preferences are considered. Abilities, together with well-being, should be considered, to ensure that both health aspects of Tengland’s theory are addressed.

Considering the complexity of sleep, using a range of research methods is important to capture the various aspects of the phenomenon. These subjective reports from parents contribute to more knowledge on sleep and the numerous ways it influences the abilities and well-being of a child with ADHD. Further research is needed on the children’s experiences of the value of sleep and how having a child with ADHD and sleep problems affects the parents’ own health and well-being. Another perspective that would be valuable to explore further is that of teachers, and how they experience the abilities and well-being of children with ADHD related to sleep problems.

## Figures and Tables

**Figure 1 ijerph-19-15099-f001:**
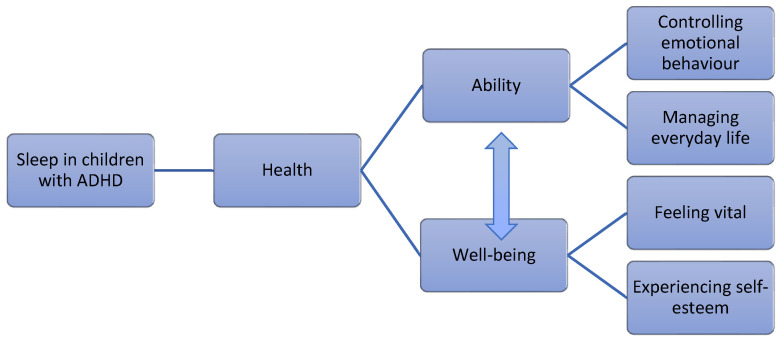
The results describe parents’ experiences of direct and indirect implications of sleep quality on the health of children with ADHD. This figure is based on Tengland’s [26] two-dimensional theory of health, where ability and well-being form the two main categories, followed by four subcategories.

**Table 1 ijerph-19-15099-t001:** Sociodemographic data of parent participants in the study experiencing how sleep influenced the health of their children with ADHD (*n* = 21).

Variable	Parents (*n* = 21)
**Sex,** female/male (*n*)	16/5
**Age in years,** median (range)	39 (32–48)
**Civil Status,** co-habiting/living alone (*n*)	17/4
**Educational level,** lower secondary/upper secondary/university (*n*)	2/8/11
**Employment,** full time/part-time/unemployed/sick leave (*n*)	12/7/1/1
**Native-born/foreign-born** (*n*)	19/2
**Place of residence,** city/countryside (*n*)	7/14
**Age of the child in years,** median (range)	9 (6–13)
**Sex of the child,** female/male (*n*)	8/11

## Data Availability

Not applicable. The data will not be shared because the ethics approval for the study requires that data files and the transcribed interviews are kept in locked files, accessible only to the researchers.

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
