# Peer review of "Parents’ Experiences of Direct and Indirect Implications of Sleep Quality on the Health of Children with ADHD: A Qualitative Study"

_ijerph, 2022, doi:10.3390/ijerph192215099_

Round 1
Reviewer 1 Report
I thank the authors for the opportunity to review this interesting article, it is an interesting article. However, please bear in mind the following recommendations and I would appreciate your answers to the questions raised:
• It would be interesting to define the type of qualitative study carried out (e.g., ethnography, grounded theory, case study, phenomenology, narrative research)
• Children aged 6-13 are included in the study. Don't you think that age differences between children are determining factors in activities and therapeutic approaches? These differences could affect children's sleep and the perception of parents. Please explain how you solved this.
• Define the exclusion criteria for the study
• The data collection period coincides with the first wave of Covid and the exceptional nature of the situation had important consequences in various areas of life, especially in children and their families. I believe that the parents' experience and the children's sleep disturbances were profoundly affected and are not representative of ordinary situations. Please detail how you controlled this situation.
• They detail that 4 interviews were conducted as a group. Don't you think that this could have conditioned the discourse due to the presence of the couple? Why did you choose to conduct some individual interviews and others in pairs? I believe that the way in which the interviews were conducted could have conditioned the discourse of the participants.
• Give more details of how the data was collected, during the data collection period there were restrictions and social isolation. Were the interviews conducted in person? To what extent did the pandemic affect data collection?
• Detail the number of units of meaning identified in the analysis for each of the subcategories.
• In the description of the results you speak of school activity, please give details of the school context during the first wave of covid, since in many centers activity was not resumed until the following academic year. Similarly, I am surprised that this situation is not referred to throughout the results due to its influence on the routines of children and parents.
Thank you very much. All the best
Author Response
Response to editor and reviewers
We thank the editor and reviewers for a thorough review and helpful comments to improve the manuscript.
Please find the reviewers’ original comments and our response below (the latter using italics). In the revised manuscript, changes have been marked using track changes to identify corrections.
I thank the authors for the opportunity to review this interesting article, it is an interesting article. However, please bear in mind the following recommendations and I would appreciate your answers to the questions raised:
- It would be interesting to define the type of qualitative study carried out (e.g., ethnography, grounded theory, case study, phenomenology, narrative research)
Answer: Thank you for the comment. We have added more information under the Study design, page 2 lines 93-97.
- Children aged 6-13 are included in the study. Don't you think that age differences between children are determining factors in activities and therapeutic approaches? These differences could affect children's sleep and the perception of parents. Please explain how you solved this.
Answer: Thank you for the comment. We have added information under Methodological considerations describing our thoughts on this matter, page 11 lines 489-493.
- Define the exclusion criteria for the study
Answer: Thank you for the comment. Parents with insufficient Swedish to take part in an interview were excluded. We have added this information in the Participants section on page 3, lines 109-110.
- The data collection period coincides with the first wave of Covid and the exceptional nature of the situation had important consequences in various areas of life, especially in children and their families. I believe that the parents' experience and the children's sleep disturbances were profoundly affected and are not representative of ordinary situations. Please detail how you controlled this situation.
Answer: Thank you for raising this issue. We have added information about the Swedish strategy during the pandemic under method and data collection on page 3, lines 116-120 and Methodological considerations. Page 11, lines 506-514.
- They detail that 4 interviews were conducted as a group. Don't you think that this could have conditioned the discourse due to the presence of the couple? Why did you choose to conduct some individual interviews and others in pairs? I believe that the way in which the interviews were conducted could have conditioned the discourse of the participants.
Answer: Thank you for the comment. Parents were allowed to choose individual or pair interviews. We have added more information in Methodological considerations Page 11, lines 494-497.
- Give more details of how the data was collected, during the data collection period there were restrictions and social isolation. Were the interviews conducted in person? To what extent did the pandemic affect data collection?
Answer: Thank you for the comment. We have added information on the situation in Sweden during the pandemic in the sections Data collection (page 3, lines 116-120) and Methodological considerations (page 11, lines 506-514).
- Detail the number of units of meaning identified in the analysis for each of the subcategories.
Answer: Thank you for the comment. We have added this information under Data analysis. Page 4, lines 156-158.
- In the description of the results you speak of school activity, please give details of the school context during the first wave of covid, since in many centers activity was not resumed until the following academic year. Similarly, I am surprised that this situation is not referred to throughout the results due to its influence on the routines of children and parents.
Answer: Thank you for the comment. We have added information about the Swedish strategy during the pandemic in Methodological considerations. Page 11 lines 506-547.
Thank you very much. All the best
Reviewer 2 Report
The present study has interesting insights into the field of sleep research in individuals with ADHD. Personally I have some doubts about it:
1. Why did Authors choose not to use internationally validated tests such as SDSC or PSQ?
2. No mention of the relationship between respiratory disorders in sleep and ADHD is present in the text.
3. The mention of the use of melatonin in the sleep disorders of subjects with ADHD appears pleonastic and out of context.
4. I believe that the purpose of the study needs to be better clarified.
Author Response
Response to editor and reviewers
We thank the editor and reviewers for a thorough review and helpful comments to improve the manuscript.
Please find the reviewers’ original comments and our response below (the latter using italics). In the revised manuscript, changes have been marked using track changes to identify corrections.
Review 2: Comments and Suggestions for Authors
The present study has interesting insights into the field of sleep research in individuals with ADHD. Personally I have some doubts about it:
- Why did Authors choose not to use internationally validated tests such as SDSC or PSQ?
Answer: Thank you for the comment. Since this is a qualitative study with in-depth interviews to understand the participants’ experiences and perceptions of the phenomenon “children’s sleep” we didn’t use any validated tests. In a companion randomized controlled study in manuscript we used actigraph to measure the childrens’ sleep as well as validated tests.
- No mention of the relationship between respiratory disorders in sleep and ADHD is present in the text.
Answer: Thank you for raising this issue. We agree that a more rich description of different types of sleep disorders among children with ADHD may strengthen the manuscript. An additional text about different sleep disorders, including sleep-related breathing disorders, has been added in the introduction. Pages 1-2, lines 44-49.
- The mention of the use of melatonin in the sleep disorders of subjects with ADHD appears pleonastic and out of context.
Answer: Thank you for the comment. We believe that describing pharmacological and non-pharmacological interventions is relevant in the introduction, following the previous section that describes prevalence, different types and origins of sleep problems. Also, the aim of the study is to explore and gain more knowledge on the sleep and how it influences health in children with ADHD, to develop and implement sleep interventions, and it is therefor of value to know what sort of interventions that are currently offerered to children with ADHD. We have removed one sentence about melatonin relating to the Swedish context. Page 2, lines 67-70.
- I believe that the purpose of the study needs to be better clarified.
Answer: Thank you for the comment. We have clarified the purpose of the study in the introduction. Page 2 lines 86-90.
Round 2
Reviewer 1 Report
After introducing the suggested changes, I consider that the article can be published